# MMedPO: Aligning Medical Vision-Language Models with Clinical-Aware Multimodal Preference Optimization

Kangyu Zhu [* 1 2]  Peng Xia [* 1]  Yun Li [1]  Hongtu Zhu [1]  Sheng Wang [3]  Huaxiu Yao [1]

## Abstract

The advancement of Large Vision-Language Models (LVLMs) has propelled their application in medicine. However, Medical LVLMs (Med-LVLMs) encounter factuality issues due to modality misalignment, where the models prioritize textual knowledge over visual input, causing hallucinations that conflict with medical images. Previous attempts on preference optimization have inadequately mitigated clinical relevance in preference data, making these samples easily distinguishable and reducing alignment effectiveness. To address this challenge, we propose MMedPO, a novel multimodal medical preference optimization approach that considers the clinical relevance of preference samples to enhance Med-LVLM alignment. MMedPO curates multimodal preference data by introducing two types of dispreference: (1) plausible hallucinations injected through target Med-LVLMs or GPT-4o to produce medically inaccurate responses, and (2) lesion region neglect achieved through local lesion-noising, disrupting visual understanding of critical areas. We then calculate clinical relevance for each sample based on scores from Med-LLMs and visual tools, and integrate these scores into the preference optimization process as weights, enabling effective alignment. Our experiments demonstrate that MMedPO significantly enhances factual accuracy, achieving improvements over existing baseline methods by averaging 14.2% and 51.7% across the Med-VQA and report generation tasks. Our code are available in https://github.com/aiming-lab/MMedPO.

---

*Equal Contribution. Work was done during Kangyu Zhu's internship at UNC. [1]UNC Chapel-Hill [2]Brown University [3]University of Washington. Correspondence to: Kangyu Zhu <kangyu@unc.edu>, Peng Xia <pxia@cs.unc.edu>, Huaxiu Yao <huaxiu@cs.unc.edu>.

*Proceedings of the 42nd International Conference on Machine Learning*, Vancouver, Canada. PMLR 267, 2025. Copyright 2025 by the author(s).

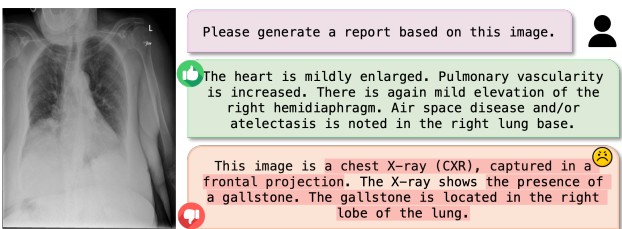

*Figure 1.* An illustration of preference data pair. The dispreferred response contains nonfactual and clinically meaningless content.

## 1. Introduction

Artificial intelligence is increasingly being applied in the medical field (Tăuţan et al., 2021; Wang et al., 2019; Ye et al., 2021; Tu et al., 2024; Xia et al., 2024c; Wang et al., 2025; Hu et al., 2024; 2023; Li et al., 2024), including areas such as disease diagnosis and treatment planning. With the recent surge in popularity of Large Vision-Language Models (LVLMs) (Liu et al., 2024b;a; Zhu et al., 2023), Medical LVLMs (Med-LVLMs) have begun to develop rapidly, drawing significant attention (Li et al., 2023a; Moor et al., 2023; Zhang et al., 2023; Wu et al., 2023c; Xia et al., 2024f;e). However, these models still face the challenge of factuality, which is largely due to modality misalignment issues (Cui et al., 2023; Zhou et al., 2024a; Sun et al., 2024). Models with poor modality alignment tends to prioritize the textual knowledge learned during training over the actual visual input. As a result, Med-LVLMs often produce hallucinations, generating text that appears coherent but contradicts the information in the corresponding medical image (Xia et al., 2024a; Royer et al., 2024).

To tackle this issue, several studies have employed preference optimization on Med-LVLMs, aiming to improve alignment between medical image and text modalities with factuality improvement (Hein et al., 2024; Sun et al., 2024; Banerjee et al., 2024). However, these methods simply leverage the preference data generation process used for aligning general LVLMs on natural images, overlooking the clinical relevance of the generated preference samples. Consequently, these preference samples become relatively easily distinguishable, reducing their effectiveness in align-

ing Med-LVLMs. Clinical relevance can be considered from two perspectives. First, in these preference samples, it is essential that both preferred and dispreferred responses are clinically meaningful; if dispreferred responses lack clinical relevance, Med-LVLMs can easily distinguish them, diminishing the sample's effectiveness. For instance, a dispreferred response such as "*a gallstone in the right lobe of the lung...*" reflects a clear factual error with limited clinical relevance (Tu et al., 2024). Second, when improving alignment between the generated medical response and the input medical image, focused attention on local lesion areas is essential for accurate medical image understanding. Correcting dispreferred responses that arise from overlooking these lesion regions is crucial for achieving more precise medical alignment.

To address this challenge, we introduce **MMedPO**, a novel **M**ultimodal **Med**ical **P**reference **O**ptimization approach designed to quantify preference sample importance based on clinical relevance, enabling more effective preference optimization in Med-LVLMs. In MMedPO, we first curate multimodal medical preference data using two strategies: (1) introducing dispreference by leveraging target Med-LVLMs (Li et al., 2023a) or GPT-4o (OpenAI, 2023) to inject plausible hallucinations into responses, ensuring dispreferred outputs contain evident medical inaccuracies, such as incorrect imaging interpretations, misleading descriptions, or inaccurate diagnoses; and (2) provoking dispreference by neglecting lesion regions through a visual tool-guided local lesion-noising process, which disrupts the model's understanding of these areas, leading to responses that overlook critical regions, thus being marked as dispreferred. We then quantify each preference sample's clinical significance by formulating sample importance scores, which integrate (1) clinical significance scores of dispreferred responses, evaluated by a multiple Med-LLMs collaboration process, and (2) confidence scores from visual tools to assess lesion region detection accuracy. These sample importance scores are then feed into a preference optimization process, enabling more effective alignment based on the clinical relevance of each preference sample.

The primary contribution of this paper is the introduction of MMedPO, aiming to quantify the clinical significance of curated preference samples to achieve more effective alignment and enhance factual accuracy in Med-LVLMs. Empirical results on two Medical Visual Question Answering (Med-VQA) (Lau et al., 2018; Liu et al., 2021) and two report generation datasets (Johnson et al., 2020; Demner-Fushman et al., 2016) demonstrate that MMedPO substantially improves the factual accuracy of Med-LVLMs, achieving significant gains over the best previous preference optimization methods, with improvements of 14.2% and 51.7% on the Med-VQA and report generation tasks, respectively.

## 2. Preliminaries

### 2.1. Medical Large Vision Language Models

Medical Large Vision-Language Models (Med-LVLMs) are advanced architectures primarily comprising a Large Language Model (LLM) integrated with a specialized visual module. The visual module analyzes medical images to extract relevant information, transforming it into a representation compatible with the LLM's processing capabilities. Given a medical image $x_v$ and a clinical query $x_t$, the combined input is represented as $x = (x_v, x_t)$. The model then autoregressively predicts the probability distribution of the next token in the sequence, leveraging the multimodal input. The text output generated by the model is denoted as $y$.

### 2.2. Preference Optimization

Preference optimization has proven highly effective in fine-tuning LLMs (Rafailov et al., 2023; Bai et al., 2022), leading to a significant alignment between model behavior and target objectives. In preference optimization, given an input $x$, the language model policy $\pi_\theta$ generates a conditional distribution $\pi_\theta(y \mid x)$, where $y$ represents the output text response. One of the notable methods, Direct Preference Optimization (DPO) (Rafailov et al., 2023), leverages preference data to facilitate alignment within LLMs. The preference dataset is defined as $\mathcal{D} = \{(x^{(i)}, y_w^{(i)}, y_l^{(i)})\}_{i=1}^N$, where $y_w^{(i)}$ denotes the preferred response and $y_l^{(i)}$ the dispreferred response for a given input $x$. The probability of preferring $y_w$ over $y_l$ is modeled as $p(y_w \succ y_l) = \sigma(r(x, y_w) - r(x, y_l))$, with $\sigma(\cdot)$ representing the sigmoid function. In DPO, the optimization process is expressed as a following loss computed over the preference data:

$$\mathcal{L}_{DPO}(\pi_\theta; \pi_{\text{ref}}) = -\mathbb{E}_{(x, y_w, y_l) \sim \mathcal{D}}$$
$$\left[ \log \sigma \left( \alpha \log \frac{\pi_\theta(y_w|x)}{\pi_{\text{ref}}(y_w|x)} - \alpha \log \frac{\pi_\theta(y_l|x)}{\pi_{\text{ref}}(y_l|x)} \right) \right]. \quad (1)$$

where $\pi_\theta$ represents the reference policy, which is the LLM fine-tuned through supervised fine-tuning.

## 3. Multimodal Medical Preference Optimization (MMedPO)

In this section, we propose MMedPO, a clinical-aware multimodal preference optimization method to address modality misalignment challenges in Med-LVLMs, which consists of three steps and the entire framework is illustrated in Figure 2. Firstly, we use the target Med-LVLM or GPT along with medical visual tools to jointly construct medical multimodal preference data. Second, we evaluate the clinical relevance of each preference sample using a collaborative process with multiple Med-LLMs and confidence scores from medical visual tools for lesion region detection. Lastly, the normalized clinical relevance scores are integrated into the preference

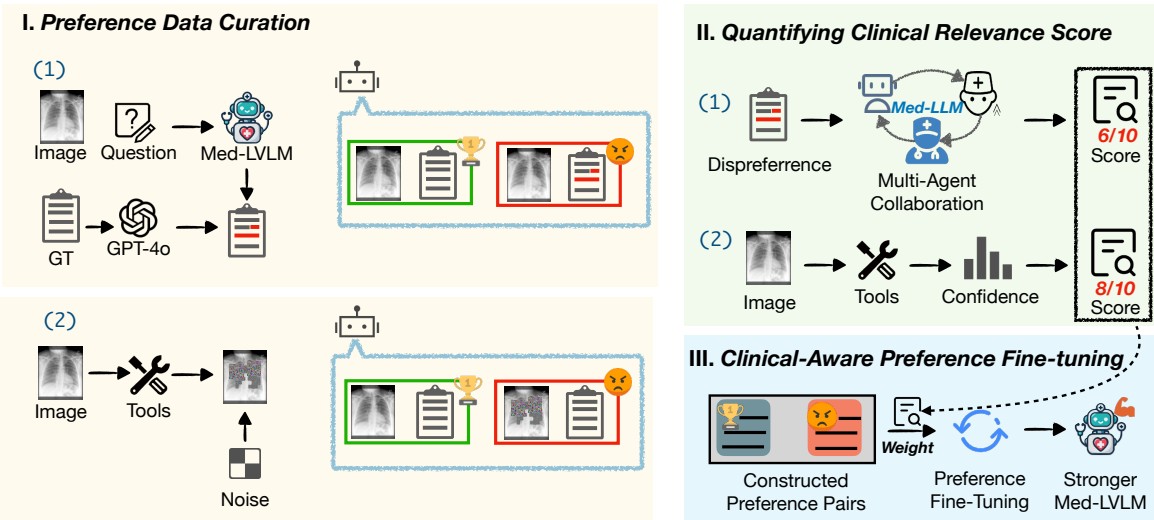

*Figure 2.* The overview of MMedPO outlines a comprehensive framework consisting of multimodal preference data curation, a quantified preference scoring module, and clinical-aware preference optimization. For data curation, the hallucinated text response and localized noisy images are joint constructed as preference data. Then the clinical relevance score is obtained through a multi-agent collaboration system and visual tools. Finally, these scores, serve as weights for the clinical-aware preference optimization.

optimization process to achieve clinical-aware preference optimization. We detail these steps as follows:

### 3.1. Preference Data Curation

In the first step, our goal is to construct a high-quality, medical-specific multimodal preference dataset using two strategies: (1) introducing dispreference by using target Med-LVLMs or GPT-4o (OpenAI, 2023) to inject hallucinations into medical responses, ensuring that dispreferred responses include significant medical inaccuracies; (2) provoking dispreference by neglecting lesion regions through a medical visual tool-augmented local lesion-noising process, resulting in dispreferred responses that overlook critical regions. We detail both strategies as follows:

**Generating Hallucinated Medical Responses**. In the first strategy, we aim to generate a hallucinated medical response, designated as the dispreferred response, while the ground truth serves as the preferred response. To achieve this, we first perform multiple rounds of sampling on the target Med-LVLMs $\mathcal{M}(\cdot)$ to collect a set of potential hallucinated responses. We then use GPT-4o to evaluate all candidate responses and select the response with the highest level of hallucination, displaying clear conflicts with the ground truth. If none of the candidates exhibit significant hallucinations, we use GPT-4o to generate a new hallucinated response based on ground truth to ensure that dispreference contain factual inaccuracies, such as incorrect imaging interpretations, misleading condition descriptions, or erroneous diagnoses. The preference pairs constructed using this strategy are denoted as $\mathcal{D}_t$.

**Adding Noise to Localized Lesion Region**. To improve the

alignment between generated medical responses and input medical images, concentrated attention on localized lesion areas is vital for accurate interpretation. Thus, we construct dispreferred response that stem from neglecting these lesion regions. Specifically, we leverage a medical visual tool (e.g., MedKLIP (Wu et al., 2023b)) $\mathcal{T}(\cdot)$, to predict disease-related local regions $h = \mathcal{T}(x_v)$ for each medical image $x_v$. We then introduce noise into these detected localized lesion regions within the original image. The noise step is defined as $k$, and the noised image at step $k$ can be expressed as follows:

$$x_v^* = \sqrt{\bar{\xi}_k} \cdot (x_v \odot h) + \sqrt{1 - \bar{\xi}_k} \cdot (\epsilon \odot h) + (x_v \odot (1-h)), \quad (2)$$

where $\bar{\xi}_t = \prod_{i=0}^{k} \xi_i$ and $\xi_k \in (0, 1)$ are hyperparameters. In this approach, the original image $x_v$ paired with the ground truth $y$ is considered preferred, while the image with localized noise $x_k$ paired with the same ground truth $y$ is regarded as dispreferred. The preference data constructed using this strategy is denoted as $\mathcal{D}_v$.

Finally, we merge the two preference sets generated by the above two strategies and denote the preference dataset as $\mathcal{D}_o = \mathcal{D}_t \cup \mathcal{D}_v = \{x^{(i)}, x^{*(i)}, y_w^{(i)}, y_l^{(i)}\}_{i=1}^N$, where $x^{(i)}$ and $x^{*(i)}$ denote the normal and noisy input, $y_w^{(i)}, y_l^{(i)}$ represent preferred and dispreferred responses, respectively.

### 3.2. Quantified Clinical Relevance Score

After obtaining multimodal medical preference data, we will quantify the clinical relevance of each preference sample to drive effective optimization. Our hypothesis is that responses with higher clinical relevance are more valuable for preference optimization, while low-quality responses,

in turn, reduce the effectiveness of optimization. We will explain in detail how clinical relevance is calculated below.

### 3.2.1. CLINICAL RELEVANCE SCORES FOR DISPREFERRED MEDICAL RESPONSES

For samples generated by the target Med-LVLM and GPT-4o (i.e., samples in $\mathcal{D}_t$), we evaluate the clinical relevance of the dispreferred response based solely on the model's internal medical knowledge, without the need for visual input (Tian et al., 2024; Thirunavukarasu et al., 2023). Including medical images for this evaluation is unnecessary and may even hinder the process. Therefore, we rely on Med-LLMs with high levels of medical expertise to assess the clinical relevance of these text responses. Moreover, relying on a single Med-LLM for evaluating clinical relevance may introduce bias and result in unreliable assessments (Chan et al.). To address this, we implement a multi-agent collaboration system comprising multiple Med-LLMs, each with varying levels of medical expertise. These Med-LLMs collaborate through a structured debating process to reach a consensus on clinical relevance scores, thereby improving the reliability of clinical relevance evaluations.

Specifically, for each Med-LLM $\mathcal{G}_i$, where $0 < i \leq g$ and $g$ represents the total number of Med-LLMs, the objective of the multi-agent collaborative system is to establish consensus on the clinical relevance score across all agents (i.e., Med-LLMs). This process comprises $r$ rounds. In each round, each Med-LLM evaluates the clinical relevance score passed from the previous Med-LLM. The process begins with the first Med-LLM, $\mathcal{G}_1$, which evaluates a dispreferred response $y_l$, generating a clinical relevance score $s_1 = \mathcal{G}_1(y_l)$ and recording it in the score history $S$. Subsequently, each following Med-LLM $\mathcal{G}_i$ retrieves the prior scores $s_{i-1}$ and determines whether to agree. If a Med-LLM concurs, it adopts $s_{i-1}$ as its clinical relevance score $s_i$; otherwise, it generates a new score as $s_i$. This process continues until all Med-LLMs reach consensus and produce a final score. To prevent excessive evaluations, a threshold limits the number of evaluation rounds. If this threshold is reached before consensus, the final score is defined as the average of the scores in the history: $\hat{s} = \frac{\sum_{i=1}^{|S|} s_i}{|S|}$, ensuring efficient consensus that reflects clinical relevance, where $|S|$ represents the total number of scores.

### 3.2.2. CONFIDENCE SCORES FOR LOCALIZED LESION REGIONS FROM VISUAL TOOLS

For preference data in $\mathcal{D}_v$, distinct noisy regions correspond to disease-related lesion areas. Introducing noise into images to generate dispreferred responses for preference comparison can improve the visual understanding of Med-LVLMs (Zhou et al., 2024a; Zhao et al., 2023; Wang et al., 2024a). Emphasizing lesions associated with the dis-

ease through noise can further enhance the model's focus on these critical areas. However, if noisy regions are inaccurately defined, the reliability of these samples decreases, potentially impacting the model performance. Therefore, quantifying the accuracy of critical lesion detection to represent sample importance during optimization is importance. To achieve this, we use the confidence scores $s$ from visual tools that generate heatmaps of local regions as an indicator of clinical relevance. We assign different clinical relevance scores to preference pairs based on the confidence scores provided by visual tools for lesion detection.

### 3.3. Clinical-Aware Preference Fine-tuning

Following the previous steps, we construct multimodal medical preference data and assign a quantified clinical relevance score to each preference sample. During preference optimization, we treat this score as the sample weight representing the contribution of each preference data pair to the overall objective. To prevent underfitting caused by an excessively small overall loss, we apply a normalization strategy, mapping the scores to a fixed range while maintaining their mean and variance. Specifically, for each clinical relevance score $s$, the normalized score $s'$ is calculated as: $s' = \frac{(s-\mu)}{\sigma}$, then we clip $s'$ to values of $[\alpha, \beta]$. Here $\alpha$ and $\beta$ denote the predefined upper and lower bounds for the normalized score, and $\mu$ and $\sigma$ represent the mean and variance of the original scores, respectively. After obtaining the normalized clinical relevance score, we fine-tune the Med-LVLM using a weighted DPO. Following Eqn. 3, the adjusted loss with clinical relevance as sample weights is calculated as follows:

$$\mathcal{L}_{mmedpo} = -\mathbb{E}_{(x,x^*,y_w,y_l,s')\sim\mathcal{D}_o} \left[ s' \log \sigma \left( \alpha \log \frac{\pi_\theta(y_w|x)}{\pi_o(y_w|x)} - \alpha \log \frac{\pi_\theta(y_l|x^*)}{\pi_o(y_l|x^*)} \right) \right]. \quad (3)$$

## 4. Experiment

In this section, we evaluate the effectiveness of MMedPO to answer the following questions: (1) Can MMedPO enhance the factual accuracy of Med-LVLMs compared to other alignment baselines? (2) How does each individual component of the framework contribute to overall performance? (3) Can MMedPO be compatible with different Med-LVLM architectures? (4) Does MMedPO improve Med-LVLMs' responses in terms of clinical relevance?

### 4.1. Experimental Setups

**Evaluation Datasets.** To verify the effectiveness of MMedPO in improving factuality, we utilize four medical datasets: two medical VQA datasets, i.e., VQA-RAD (Lau et al., 2018) and SLAKE (Liu et al., 2021), and two report generation datasets, i.e., MIMIC-CXR (Johnson et al., 2020)

---

**Algorithm 1:** Multimodal Medical Preference Optimization (**MMedPO**)

**Input:** $\mathcal{D} = \{x_v^{(i)}, x_t^{(i)}, y^{(i)}\}_{i=1}^N$: Dataset; $\mathcal{M}(\cdot,\cdot)$: Med-LVLM; $\mathcal{T}(\cdot)$: Visual Tool; $\mathcal{G}(\cdot)$: Med-LLM; $\mathcal{N}(\cdot,\cdot)$: Localized Nosiy Process; $\mathcal{Z}(\cdot)$: Normalization.

**Output:** $\pi_\theta$: Parameters of the Med-LVLM.

1   Initialize $\mathcal{D}_o$ with an empty set
2   **foreach** $(x_v, x_t, y) \in \mathcal{D}$ **do**
3      ▷ *Preference Data Curation*
4      Generate responses of the Med-LVLM $a \leftarrow \mathcal{M}(x_v, x_t)$
5      Select the dispreferred response $y_l \leftarrow \text{GPT}(a, y)$
6      ▷ *Quantify the Clinical Relevance*
7      Quatify the clinical relevance using Med-LLMs $s_t \leftarrow \mathcal{G}(y_l)$
8      Put $\{x_v, y, y_l, s_t\}$ into $\mathcal{D}_o$;
9      Obtain the heatmap of lesion region $h \leftarrow \mathcal{T}(x_v)$
10     Save the confidence score from visual tool $s_v \leftarrow P(h|x_v)$
11     Add noise to the localized region $x_v^* \leftarrow \mathcal{N}(x_v, h)$
12     Put $\{x_v, x_v^*, y, s_v\}$ into $\mathcal{D}_o$;
13 ▷ *Clinical Preference Optimization*
14 **foreach** $(x, x^*, y_w, y_l, s) \in \mathcal{D}_o$ **do**
15     Normalize the score $s' \leftarrow \mathcal{Z}(s)$
16     Update $\pi_\theta$ through Eq. (3)

---

and IU-Xray (Demner-Fushman et al., 2016).

**Implementation Details.** We utilize LLaVA-Med-1.5 7B (Li et al., 2023a) as the base model. During the preference optimization stage, we apply LoRA fine-tuning (Hu et al., 2021), with a batch size of 4, a learning rate of 1e-7, and train for 3 epochs. For curating preference data, we use GPT-4o to evaluate and generate dispreferred responses. In the multi-agent collaboration system, multiple Med-LLMs, including LLaMA3-Med42-7B (Christophe et al., 2024), LLaMA3-Med42-70B, BioMistral-7B (Labrak et al., 2024), are used to evaluate the relevance scores for the preference data. See Appendix B for more details.

**Baselines.** We compare MMedPO with Direct Preference Optimization (DPO) (Rafailov et al., 2023) and its variants, including the self-rewarding method (Yuan et al., 2024) and STLLaVA-Med (Sun et al., 2024). In the self-rewarding method, the model generates its own responses to form preference pairs, while STLLaVA-Med further refines the preference selection process using GPT-4o and apply it in Med-LVLMs. We further compare three VLM preference fine-tuning methods originally designed for natural images: POVID (Zhou et al., 2024a), FiSAO (Cui et al., 2024), and SIMA (Wang et al., 2024b). Additionally, we evaluate MMedPO and all baselines on models that have undergone supervised fine-tuning (SFT) with the corresponding datasets and compare their performance. Please see more details in Appendix C.

**Evaluation Metrics.** For Med-VQA task, we use accuracy and recall for both closed-ended and open-ended questions. For the report generation task, we use BLEU Score (Papineni et al., 2002), ROUGE-L (Lin, 2004) and METEOR (Banerjee & Lavie, 2005) as the metrics.

### 4.2. Main Results

In this section, we present a comprehensive comparison of MMedPO with baseline methods.

**Comparison with Baseline Methods.** As shown in Table 1, we evaluate our model's performance against the original LLaVA-Med-1.5 and several preference optimization baselines. MMedPO demonstrates superior performance across both Medical VQA and report generation tasks. Specifically, for Med-VQA task, MMedPO significantly outperforms the best baseline (i.e., original DPO) by an average of 15.8% and 10.3% across the open-ended and closed-ended questions, respectively. We also observe that the overall performance improvement on open-ended questions is greater than that on closed-ended questions, indicating that MMedPO is particularly effective in guiding accurate open-ended generation. Additionally, MMedPO exhibits superior performance on the report generation task, surpassing the best baseline by 61.9% and 26.0% on IU-Xray and MIMIC-CXR, respectively. This demonstrates that, by constructing a multimodal preference dataset and assigning quantified clinical relevance scores to measure sample importance, MMedPO ensures that clinical relevance is fully considered during the preference optimization process, resulting in more accurate and clinically meaningful responses.

**Comparison with Baseline Methods Enhanced by SFT.** To demonstrate the compatibility of our approach with other training methods, we conduct experiments by applying MMedPO and other baseline methods to SFT. As shown in Table 1, MMedPO consistently outperforms the SFT baseline across all four datasets, with an average improvement of 14.2%. When compared to other baselines applied to SFT, MMedPO achieves significantly better performance, with an average improvement of 10.5%. These results further corroborate the effectiveness and compatibility of our approach, demonstrating its ability to integrate seamlessly with other training techniques to enhance model alignment.

### 4.3. Quantitative Analysis

In this section, we first conduct ablation study to analyze the effectiveness of each strategy and component in MMedPO for enhancing factual accuracy. Then, we evaluate the model's compatibility with different backbones. We further explore how our approach improves Med-LVLMs' responses in terms of clinical significance and visual understanding.

*Table 1.* Performance comparison on medical VQA and report generation tasks covering SLAKE, VQA-RAD, and IU-Xray datasets. For open-ended questions, we report recall (Open); for closed-ended questions, accuracy (Closed). The BLEU score denotes the average of BLEU-1/2/3/4. +SFT indicates that the model is first fine-tuned with SFT before applying the corresponding baselines. The best results and second best results are highlighted in red and blue, respectively.

| Models | SLAKE | | VQA-RAD | | IU-Xray | | | MIMIC-CXR | | |
| | Open | Closed | Open | Closed | BLEU | ROUGE-L | METEOR | BLEU | ROUGE-L | METEOR |
| --- | --- | --- | --- | --- | --- | --- | --- | --- | --- | --- |
| LLaVA-Med v1.5 | 44.26 | 61.30 | 29.24 | 63.97 | 14.56 | 10.31 | 10.95 | 10.25 | 9.38 | 7.71 |
| + Self-Rewarding | 42.63 | 61.30 | 33.29 | 64.17 | 14.20 | 10.38 | 10.52 | 10.78 | 9.27 | 7.73 |
| + DPO | 49.30 | 62.02 | 29.76 | 64.70 | 16.08 | 12.95 | 17.13 | 11.19 | 9.45 | 7.80 |
| + POVID | 52.43 | 70.35 | 31.77 | 65.07 | 20.80 | 24.33 | 30.05 | 11.21 | 9.66 | 7.84 |
| + SIMA | 51.77 | 69.10 | 31.23 | 64.80 | 17.11 | 22.87 | 29.10 | 11.16 | 9.58 | 7.49 |
| + FiSAO | 52.69 | 70.46 | 32.70 | 64.11 | 21.06 | 25.72 | 30.82 | 11.32 | 9.68 | 7.62 |
| + STLLaVA-Med | 48.65 | 61.75 | 30.17 | 64.38 | 16.11 | 10.58 | 10.51 | 11.11 | 9.29 | 7.72 |
| + **MMedPO(Ours)** | 53.99 | 73.08 | 36.36 | 66.54 | 23.49 | 29.52 | 34.16 | 12.85 | 11.13 | 10.03 |
| + SFT | 50.45 | 65.62 | 31.38 | 64.26 | 22.75 | 28.86 | 33.66 | 12.39 | 10.21 | 8.75 |
| + Self-Rewarding | 50.62 | 65.89 | 32.69 | 65.89 | 22.89 | 28.97 | 33.93 | 12.15 | 10.05 | 8.77 |
| + DPO | 53.50 | 69.47 | 32.88 | 64.33 | 23.07 | 29.97 | 34.89 | 12.37 | 10.38 | 9.10 |
| + POVID | 52.18 | 70.67 | 32.95 | 64.97 | 23.95 | 29.75 | 34.63 | 11.85 | 10.45 | 9.05 |
| + SIMA | 51.75 | 69.28 | 32.50 | 64.08 | 23.90 | 29.41 | 34.45 | 12.44 | 10.25 | 9.02 |
| + FiSAO | 52.80 | 70.82 | 32.94 | 65.77 | 23.57 | 29.88 | 35.01 | 12.97 | 10.69 | 9.39 |
| + STLLaVA-Med | 52.72 | 66.69 | 33.72 | 64.70 | 22.79 | 28.98 | 34.05 | 12.21 | 10.12 | 8.98 |
| + **MMedPO(Ours)** | 55.23 | 75.24 | 34.03 | 67.64 | 24.00 | 30.13 | 35.17 | 13.28 | 13.22 | 10.20 |

### 4.3.1. ABLATION STUDY

**Different Preference Curation Strategies.** To assess the impact of different preference curation strategies in MMedPO, namely generating hallucinated medical responses and adding noise to localized lesion regions, we evaluated their performance on these two components. The results, presented in Figure 3, reveal that adding noise to localized lesion regions has a more pronounced effect on open-ended generation tasks (e.g., report generation) compared to generating hallucinated medical responses. For medical VQA tasks, the performance improvements from both preference curation processes are comparable. By integrating both strategies, MMedPO achieves the best overall performance across four datasets, effectively combining their strengths to maximize performance gains.

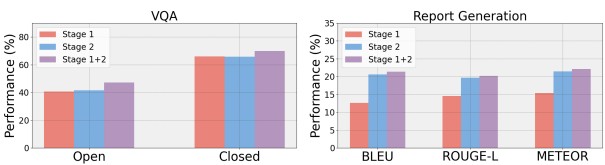

*Figure 3.* Comparison of the effectiveness of different preference curation strategies. "stage 1": generating hallucinated medical responses; "stage 2": adding noise to localized lesion regions; "stage 1+2": merged preference data. We report the average score on each dataset.

**Clinical Relevance Score.** To investigate the role of clinical relevance score as weight in the preference optimization process, we compare the results of applying this weight versus not applying it under different preference curation strategies.

*Table 2.* Comparison of performance across different datasets with and without clinical relevance score (CRS) for different preference curation strategies. Here, stage 1 and stage 2 denote generating hallucinated medical responses and adding noise to localized lesion regions, respectively. We report the average score on each dataset.

| | SLAKE | VQA-RAD | IU-Xray | MIMIC-CXR |
| --- | --- | --- | --- | --- |
| Stage 1 w/o CRS | 55.65 | 47.23 | 10.95 | 6.55 |
| Stage 1 w CRS | **57.62** | **48.67** | **15.66** | **6.58** |
| Stage 2 w/o CRS | 60.59 | 45.94 | 19.30 | 7.17 |
| Stage 2 w CRS | **60.88** | **46.97** | **25.00** | **7.24** |

The results indicate that incorporating clinical relevance scores as weights in preference optimization improves the effectiveness of fine-tuning. Specifically, as shown in Table 2, for VQA task, models utilizing clinical relevance scores as weights consistently outperform those without them, with an average improvement of 2.3%. Also, significant performance gains are observed on the report generation task, where clinical relevance scores contributed positively across different preference curation strategies, achieving a clear average margin of 18.5%. The clinical relevance scores assigned to each preference pair provide positive benefits to preference optimization, helping the Med-LVLMs generate responses that are more clinically meaningful and accurate.

### 4.3.2. MULTIPLE VS. SINGLE MED-LLM

To explore the impact of the multi-agent collaboration mechanism in generating clinical relevance scores, we conduct analytical experiments, comparing the performance using clinical relevance scores from single Med-LLM and multiple Med-LLMs. As shown in Table 3, we find that the consensus scores reached by multiple Med-LLMs positively

*Table 3.* Comparison of model performance using clinical relevance scores from single Med-LLM and multiple Med-LLMs for MMedPO. We report the average score on each dataset.

| Models | SLAKE | VQA-RAD | IU-Xray | MIMIC-CXR |
|---|---|---|---|---|
| Single-LLM | 56.09 | 48.67 | 15.67 | 6.58 |
| Multi-LLMs | **57.53** | **51.14** | **15.86** | **6.86** |

*Table 4.* Performance comparison between introducing local noise and global noise on the stage of constructing preference data by adding noise to medical images.

| Noise Location | SLAKE | VQA-RAD | IU-Xray | MIMIC-CXR |
|---|---|---|---|---|
| Global | 58.88 | 46.91 | 24.88 | 6.80 |
| Local | **59.88** | **46.98** | **25.00** | **7.24** |

contribute to performance improvement by an average of 3.6% over four datasets. This aligns with our expectations, as relying on a single Med-LLM will introduce biases. The observed improvement is driven by reduced bias through the collaborative efforts of multiple Med-LLMs, resulting in more accurate and clinically meaningful relevance evaluations. In addition, the performance gains on the Med-VQA task using multiple Med-LLMs are notably larger compared to the report generation task. This may be attributed to greater disagreement among Med-LLMs on rejected VQA answers, allowing them to benefit more from achieving consensus.

### 4.3.3. IMPACT OF LOCALIZED LESION NOISE

To evaluate the impact of localized lesion noise during the preference optimization process, we compare the performance of preference data composed of images with localized noise versus those with global noise. Global noise refers to adding noise uniformly across the entire image. As shown in Table 4, introducing localized noise consistently outperforms global noise across the four datasets. This indicates that lesion regions detected by visual tools are more prominent than the entire image. Introducing localized noise based on these regions helps the model better understand critical lesions, leading to more factually accurate responses.

### 4.3.4. COMPATIBILITY ANALYSIS

To evaluate the compatibility of our approach with different base models, particularly more powerful backbone architectures, we replace the backbone of LLaVA-Med-1.5 and conduct a series of experiments based on this configuration. Specifically, we apply our method to LLaVA-Med++ (Xie et al., 2024), which uses LLaMA-3 (Dubey et al., 2024) as language backbone and enhances its performance using a large-scale medical multimodal dataset MedTrinity-25M. As illustrated in Table 4, similar to the results obtain with LLaVA-Med-1.5, applying MMedPO leads to performance improvements across all four datasets. These findings highlight the strong compatibility and effectiveness of our ap-

proach when integrated with other powerful Med-LVLMs. MMedPO can be transferred to a wider range of base models, demonstrating strong generalizability for applications in clinical scenarios.

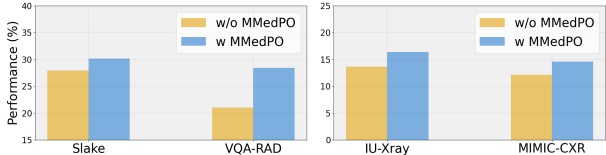

*Figure 4.* Analysis of compatibility using LLaVA-Med++ as the backbone model. Averaged metrics across datasets are presented.

### 4.4. Qualitative Analysis and Case Study

In this section, we further conduct qualitative experiments and case analyses.

#### 4.4.1. QUALITATIVE ANALYSIS

**How does MMedPO in Improving Visual Understanding?** To better understand the model's visual comprehension capability, we visualize its attention map on image tokens. As shown in Figure 5, compared to the attention map of the original model, the utilization of MMedPO significantly enhances the model's focus on visual information, particularly on critical lesion areas. This allows the model to extract sufficient information from visual inputs and improve consistency between text and images. Thus the model can reduce hallucinations and provide more accurate answers.

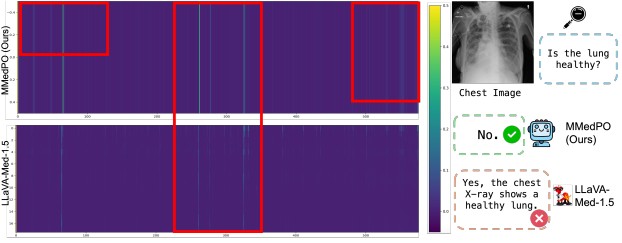

*Figure 5.* Visualization of attention map of image tokens. The red box region is labeled with the attentions that are enhanced.

**Analysis Clinical Significance of Model's Response.** Through the analysis of previous results, Med-LVLMs enhanced by MMedPO demonstrate a significant improvement in factuality accuracy. Additionally, from the clinical perspective, we aim to evaluate the clinical significance of the responses to verify the effectiveness of MMedPO in enhancing the clinical relevance of the model's outputs. As demonstrated in Figure 6, Med-LVLMs with MMedPO outperforms both the original model and the one applied with DPO. The response with MMedPO accurately capture the condition of the cardiac silhouette and rib fracture in the image, aligning with the ground truth. This also improves

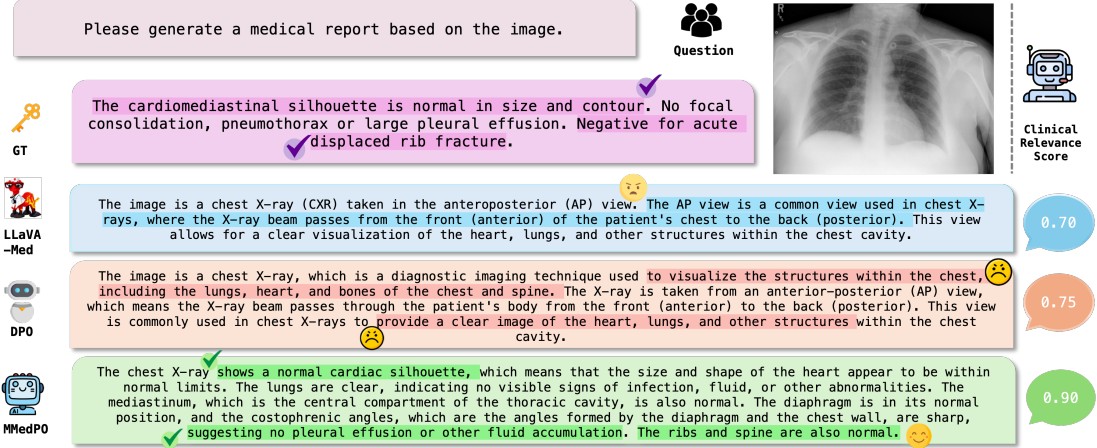

*Figure 6.* Examples demonstrating the clinical relevance of responses generated by MMedPO. Our approach not only enhances the factual accuracy but also significantly improves the clinical relevance, including various meaningful medical-level explanations.

clinical significance judged by Med-LLMs, whereas the original model and other baselines produced duplicate and clinically irrelevant content. The evaluation of response using clinical relevance from Med-LLMs quantitatively shows that MMedPO consistently achieves significantly higher clinical relevance scores.

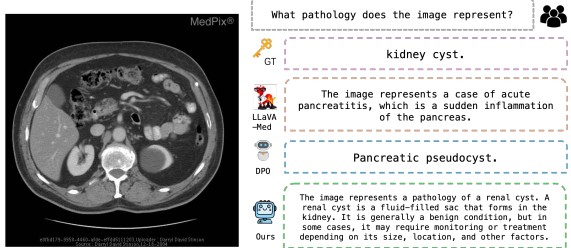

*Figure 7.* Illustration of factuality enhancement by MMedPO.

### 4.4.2. CASE STUDY

We analyze two examples from Medical VQA task to illustrate how the model fine-tuned with MMedPO reduces factuality errors. As illustrated in Figure 7, MMedPO shows improved performance in factual accuracy. In this case, when asked about pathology, MMedPO provides a more detailed response, focusing on the problem of cyst, which is similar to the ground truth, outperforming both LLaVA-Med and LLaVA-Med with DPO. This demonstrates that MMedPO effectively reduces hallucinations in Med-LVLMs, minimizing factual errors in multimodal understanding tasks.

## 5. Related Work

**Factuality Issues in Med-LVLMs**. The development of Large Vision-Language Models (LVLMs) is progressing rapidly (Liu et al., 2024a;b; Zhu et al., 2023; Bai et al., 2023; Xia et al., 2025; 2024d; Han et al., 2025; Xia et al., 2024b), which has, in turn, driven advancements in Medical Vision-Language Models (Med-LVLMs), achieving promis-

ing results in the medical field (Li et al., 2023a; Moor et al., 2023; Thawkar et al., 2023; Wu et al., 2023c). However, the current Med-LVLMs still exhibit significant factual errors (Wu et al., 2023a; Li et al., 2023b; Xia et al., 2024a; Chen et al., 2024; Jiang et al., 2024; Su et al., 2024a). For example, they often lack sufficient judgment ability for complex content, and frequently generate responses with hallucinations that contradicts the visual information provided. This issue is particularly pronounced in medical domain, as it can potentially lead to misdiagnoses or missed diagnoses. Recently, there are several benchmarks (Xia et al., 2024a; Royer et al., 2024) that highlight the factuality issues of Med-LVLMs on multiple tasks such as the visual question answering and report generation.

**Preference Optimization in Med-LVLMs.** Aligning with human preferences for large models is an effective way to address hallucination issues (Lee et al., 2024; Zhou et al., 2024a;b; Deng et al., 2024). Preference fine-tuning in LVLMs generally involves two approaches: one aligns models based on human feedback (Bai et al., 2022; Rafailov et al., 2023), while the other uses feedback generated by AI (Lee et al., 2024; Zhou et al., 2024a;b; Wang et al., 2024a; Zhou et al., 2025; Tong et al., 2025; Su et al., 2024b). Recently, the preference fine-tuning technique has also been adapted for medical imaging (Banerjee et al., 2024; Sun et al., 2024; Hein et al., 2024) by generating dispreferred responses using GPT-4 or the target Med-LVLM. Although these methods have shown promise, they neglect the clinical relevance of generated samples. In Med-LVLMs, local visual information is crucial for accurate responses, yet current approaches rarely guide the model's focus to specific lesion areas during preference fine-tuning. To tackle these issues, we incorporate quantified clinical relevance scores as weights to enhance modality alignment and introduce localized noise in medical images to construct dispreference, improving its understanding of key lesions.

# 6. Conclusion

In this work, we propose a novel clinical-aware multimodal preference optimization approach named MMedPO which considers the clinical relevance of each preference sample in preference optimization process. This method enhances Med-LVLM alignment while effectively reducing factual hallucinations. Specifically, to construct multimodal preference data, we introduce plausible hallucinations and apply local noise to critical lesion regions. Furthermore, we assign clinical relevance for data samples through Med-LLMs and visual tools, and then incorporate these scores as weights in the preference fine-tuning process. We evaluate the effectiveness of MMedPO on the Med-VQA and report generation tasks, demonstrating superior performance.

# Acknowledgement

This work is partially supported by Cisco Faculty Research Award and NIH R01AG085581, R01AG079291 and P50HD103573. The Authors acknowledge the National Artificial Intelligence Research Resource (NAIRR) Pilot, NCSA DeltaAI and OpenAI API for contributing to this research result.

# Impact Statement

The broader impact of this work lies in its potential to enhance the reliability and accuracy of AI-driven medical diagnostics by reducing hallucinations and improving visual-textual alignment in Med-LVLMs. This advancement could lead to more trustworthy AI tools in healthcare, benefiting patient outcomes. However, ethical considerations are crucial to ensure responsible deployment, prevent misuse, and avoid over-reliance on AI-generated medical advice. Future societal benefits may include reduced diagnostic errors and improved healthcare efficiency, but ongoing research and ethical oversight are essential to align these advancements with the best interests of patients and providers.

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

## A. Data

### A.1. Data Statistics

The data statistics are shown in Table 5 and Table 6. In the training datasets, the reported quantities for the two datasets in report generation represent image-report pairs, while the quantities for the two datasets in the medical VQA task represent question-answer pairs.

*Table 5.* Data statistics for the training set of four datasets under two different task settings. "Train (visual)" refers to the number of visual-only preference data, while "Train (text)" indicates the number of text-only preference data.

| Dataset | Train (visual) | Train (text) | Train (all) |
|---|---|---|---|
| IU-Xray | 2069 | 2069 | 4138 |
| MIMIC-CXR | 800 | 800 | 1600 |
| SLAKE | 4919 | 4919 | 9838 |
| VQA-RAD | 1797 | 1797 | 3594 |

*Table 6.* Data statistics of test set. #Images, #QA items and #Reports mean the number of images, QA pairs and reports, respectively.

| Dataset | #Images | #QA items | #Reports |
|---|---|---|---|
| IU-Xray | 590 | - | 590 |
| MIMIC-CXR | 200 | - | 200 |
| SLAKE | 641 | 1061 | - |
| VQA-RAD | 315 | 451 | - |

### A.2. Involved Datasets

We leverage four open-source medical vision-language datasets: MIMIC-CXR (Johnson et al., 2020), IU-Xray (Demner-Fushman et al., 2016), SLAKE (Liu et al., 2021), and VQA-RAD (Lau et al., 2018). These datasets are designed for different tasks: the first two focus on medical report generation, while the latter two are tailored for medical visual question answering.

- **IU-Xray** is a dataset that includes chest X-ray images and corresponding diagnostic reports.

- **MIMIC-CXR** is a widely accessible dataset containing chest X-ray images in DICOM format along with corresponding radiology reports.

- **SLAKE** is an English-Chinese bilingual dataset comprising 642 images and 14,028 question-answer pairs designed for training and evaluating Med-VQA systems.

- **VQA-RAD** is the first dataset manually curated by clinicians, featuring naturally occurring questions about radiology images along with corresponding reference answers.

## B. Hyperparameter Settings

For the usage of visual tools, we employ "disease" as the text description to guide MedKLIP (Wu et al., 2023b) in generating heatmaps. For multi-agent collaboration, the process is conducted over 5 rounds . During score normalization, the parameters are set as: $\alpha = 0.75$, $\beta = 1.25$, $\mu = 1$, and $\sigma^2 = 0.1$. All hyperparameters are kept consistent across the experiments to eliminate any potential bias introduced by hyperparameter tuning. All experiments are implemented using PyTorch 2.1.2 on four NVIDIA RTX A6000 GPUs, with training requiring approximately 2 to 3 hours.

## C. Involved Baselines

- **DPO** (Rafailov et al., 2023) is a fine-tuning approach designed to align large language models (LLMs) with human preferences in a stable, efficient, and computationally lightweight manner. Unlike traditional Reinforcement Learning

*Table 7.* Detailed performance comparison on report generation tasks covering IU-Xray and MIMIC-CXR datasets. BL denotes BLEU.

| Models | IU-Xray | | | | | | MIMIC-CXR | | | | | |
|---|---|---|---|---|---|---|---|---|---|---|---|---|
| | BL-1 | BL-2 | BL-3 | BL-4 | ROUGE-L | METEOR | BL-1 | BL-2 | BL-3 | BL-4 | ROUGE-L | METEOR |
| LLaVA-Med v1.5 | 38.42 | 13.40 | 4.74 | 1.67 | 10.31 | 10.95 | 29.41 | 10.19 | 3.58 | 1.26 | 9.38 | 7.71 |
| + Self-Rewarding | 38.25 | 13.17 | 3.61 | 1.08 | 10.38 | 10.52 | 29.29 | 10.32 | 3.67 | 1.30 | 9.27 | 7.73 |
| + DPO | 41.63 | 15.13 | 5.56 | 2.03 | 12.95 | 17.13 | 29.61 | 10.29 | 3.61 | 1.27 | 9.45 | 7.81 |
| + POVID | 50.84 | 20.65 | 8.38 | 3.31 | 24.33 | 30.05 | 29.68 | 10.29 | 3.61 | 1.26 | 9.66 | 7.84 |
| + SIMA | 42.67 | 16.82 | 5.98 | 2.95 | 22.87 | 29.10 | 29.58 | 10.23 | 3.59 | 1.24 | 9.58 | 7.49 |
| + FiSAO | 51.10 | 20.92 | 8.64 | 3.59 | 25.72 | 30.82 | 29.76 | 10.37 | 3.74 | 1.39 | 9.68 | 7.62 |
| + STLLaVA-Med | 42.38 | 15.27 | 5.59 | 1.20 | 10.58 | 10.51 | 29.33 | 10.27 | 3.58 | 1.27 | 9.29 | 7.72 |
| + **MMedPO** (Ours) | 55.58 | 23.93 | 10.36 | 4.40 | 29.52 | 34.16 | 33.67 | 11.91 | 4.28 | 1.54 | 11.13 | 10.03 |

*Table 8.* Detailed component ablation study on report generation tasks covering IU-Xray and MIMIC-CXR datasets. BL denotes BLEU. Here, stage 1 and stage 2 denotes generating hallucinated medical responses and adding noise to localized lesion regions, respectively.

| Models | IU-Xray | | | | | | MIMIC-CXR | | | | | |
|---|---|---|---|---|---|---|---|---|---|---|---|---|
| | BL-1 | BL-2 | BL-3 | BL-4 | ROUGE-L | METEOR | BL-1 | BL-2 | BL-3 | BL-4 | ROUGE-L | METEOR |
| + Stage 1 (Single-LLM) | 43.45 | 16.05 | 5.99 | 2.21 | 19.66 | 22.65 | 29.41 | 10.19 | 3.58 | 1.26 | 9.33 | 7.77 |
| + Stage 1 (Multi-LLMs) | 43.95 | 16.44 | 6.21 | 2.31 | 19.57 | 22.92 | 29.85 | 10.38 | 3.65 | 1.28 | 9.62 | 8.18 |
| + Stage 2 | 55.15 | 23.59 | 10.13 | 4.23 | 29.02 | 34.26 | 30.96 | 10.89 | 3.87 | 1.38 | 9.85 | 8.81 |
| + Stage 1+2 (Single-LLM) | 55.36 | 23.85 | 10.34 | 4.39 | 29.30 | 34.22 | 32.96 | 11.63 | 4.14 | 1.46 | 10.99 | 10.03 |
| + Stage 1+2 (Multi-LLMs) | 55.58 | 23.93 | 10.36 | 4.40 | 29.52 | 34.16 | 33.67 | 11.91 | 4.28 | 1.54 | 11.13 | 10.05 |

from Human Feedback (RLHF), which involves training a reward model and using reinforcement learning to maximize the reward, DPO simplifies the process by reframing the problem. It parameterizes the reward model in a way that allows the optimal policy to be derived directly through a classification loss, eliminating the need for complex sampling or extensive hyperparameter tuning during fine-tuning.

- **Self-Rewarding** (Yuan et al., 2024) is a novel approach where the language model itself acts as a judge, generating rewards via LLM-as-a-Judge prompting during training. Unlike traditional methods that rely on reward models trained from human preferences, which are limited by human performance and static design, this method enables the model to iteratively improve both its instruction-following abilities and its reward-generating quality during iterative DPO training.

- **STLLaVA-Med** (Sun et al., 2024) refines the preference selection process using GPT-4o and applies it in medical vision-language tasks. STLLaVA-Med extends the DPO approach by incorporating a self-training mechanism specifically tailored for the medical domain.

- **POVID** (Zhou et al., 2024a) addresses the hallucination problem in vision-language models by generating feedback data using AI models. It uses ground-truth instructions as preferred responses and creates dispreferred data by injecting plausible hallucinations and distorting images, integrating these strategies into an RLHF pipeline via DPO.

- **FiSAO** (Cui et al., 2024) introduces a fine-grained self-alignment optimization method that leverages the model's own visual encoder to improve vision-language alignment. By utilizing token-level feedback from the vision encoder, it enhances alignment without the need for additional external data, outperforming traditional preference tuning methods.

- **SIMA** (Wang et al., 2024b) is a framework that enhances visual and language modality alignment through self-improvement, eliminating the need for external models or data. It uses prompts from existing datasets to self-generate responses and employs an in-context self-critic mechanism with vision metrics to select optimal response pairs for preference tuning.

# D. Additional Results

In this section, we present a detailed benchmark analysis for the report generation task. Table 7 compares our method with other baseline approaches. Additionally, Tables 8 and 9 provide comprehensive component ablation results for both the Medical VQA and report generation tasks.

*Table 9.* Detailed component ablation study on SLAKE and VQA-RAD datasets for both open and closed settings. Here, stage 1 and stage 2 denotes generating hallucinated medical responses and adding noise to localized lesion regions, respectively.

| Method | SLAKE | | VQA-RAD | |
| --- | --- | --- | --- | --- |
| | Open | Close | Open | Close |
| Stage 1 (Single-LLM) | 47.99 | 64.18 | 32.27 | 65.07 |
| Stage 1 (Multi-LLMs) | 49.39 | 65.87 | 32.42 | 69.85 |
| Stage 2 | 51.25 | 68.51 | 31.09 | 62.87 |

## E. Prompts

We utilize GPT-4o to generate hallucinated responses for constructing preference data, as illustrated by the prompts in Figure 8. Subsequently, a multi-agent system comprising Med-LLMs is employed to evaluate the clinical relevance scores of these rejected responses, with the evaluation prompts shown in Figure 9.

> **Prompt for Generation of Hallucinated Response Using GPT-4o**
>
> You are provided with a ground truth report: {gt report}. Please focus on the critical lesion descriptions within the report and inject hallucinations into these key descriptions to construct a dispreferred response. Ensure that the dispreferred responses contain significant medical inaccuracies. You may refer to the following examples for guidance: <example 1>, <example 2>. Your response should consist only of the hallucination-injected response.

*Figure 8.* The instruction to GPT-4o for the rejected hallucinated answer.

> **Prompt for Med-LLMs evaluation in the collaboration process**
>
> You are given a set consisting of a question, answer, and score. The question is: {prompt}, and the answer is: {rejected_value}. The provided score(s) for this answer is: {weighted_score}. These scores were assigned by other medical large language models based on evaluation criteria that assess the clinical value of the answer, emphasizing its relevance and specificity. The evaluation also considers whether the answer directly addresses the question without relying solely on general or common knowledge. Based on the question and answer, evaluate the score yourself. If you disagree with the provided score(s), assign a new score between 0.0 and 1.0 in your response. If you agree with the provided score(s), simply copy the score or respond with "agree." Response:

*Figure 9.* The instruction to Med-LLMs for evaluating and generating clinical relevance score.

## F. More Cases

We present additional examples in Figure 10, illustrating how our method effectively reduces hallucinated errors.

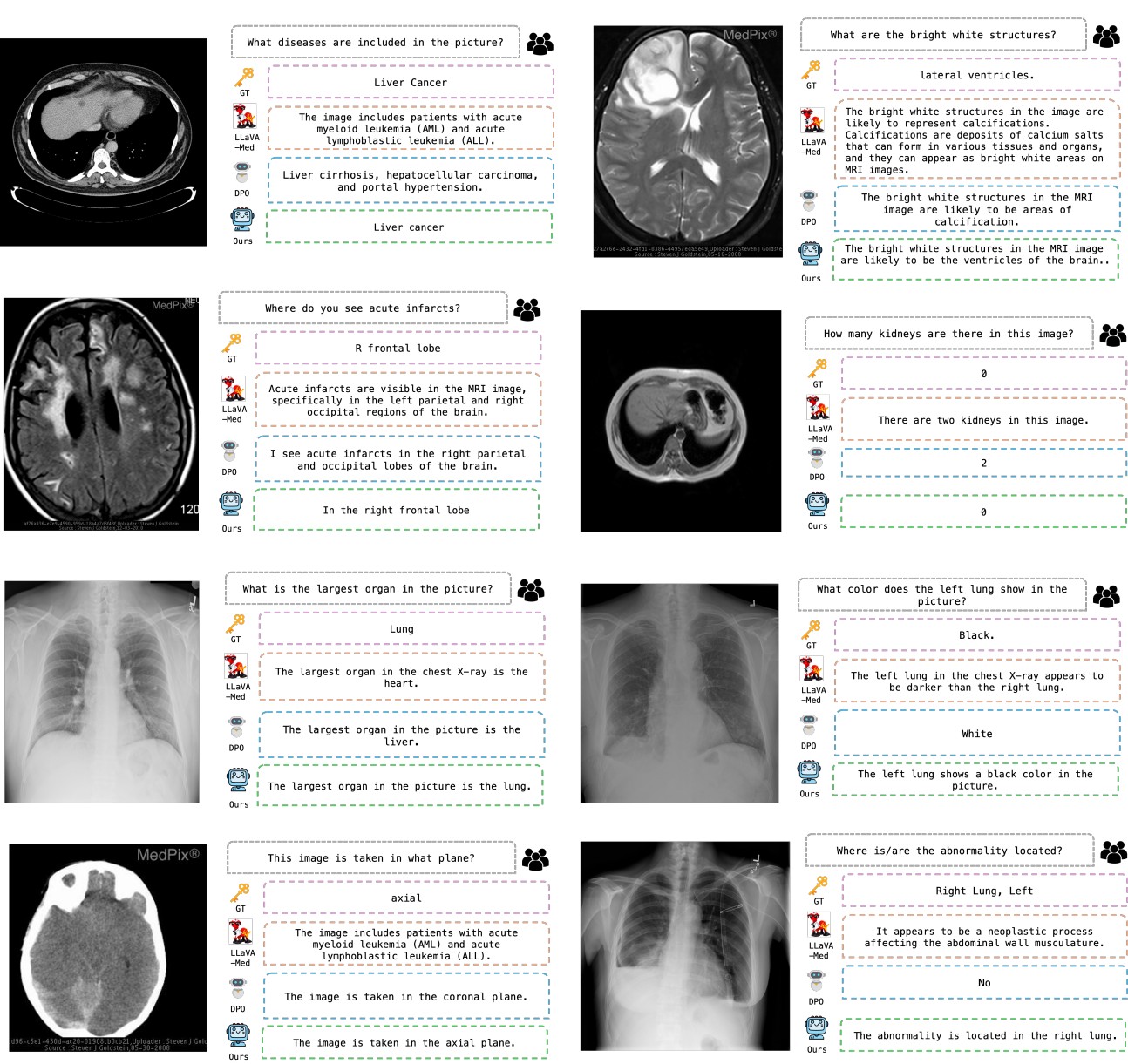

*Figure 10.* More cases that reduce hallucinated errors.

