# OpenReview forum: "MMedPO: Aligning Medical Vision-Language Models with Clinical-Aware Multimodal Preference Optimization"
_ICML.cc/2025/Conference — ICML 2025 poster_

### Official Review · Reviewer_HmNL · 2025-03-13

**Overall Recommendation:** 3

**Summary:**

This paper introduces MMedPO, a clinical-aware multimodal preference optimization approach for aligning Medical Large Vision-Language Models (Med-LVLMs). The authors leverage sentence corruption techniques from GPT-4o and a noise interaction process to generate rejected samples. They then use specialized tools to create positive answer samples from original images and employ multi-agent collaboration to evaluate the clinical relevance of the generated rejection samples. Experimental results demonstrate the effectiveness of the proposed methodology.

## update after rebuttal
The rebuttal dose help clarify my concerns and I have increased the score

**Claims And Evidence:**

The paper's methods align with the medical visual language model alignment domain. The approach of using clinical awareness for multimodal preference optimization addresses the specific challenges of medical image interpretation. The evaluation on standard medical datasets (SLAKE, VQA-RAD) is appropriate for benchmarking Med-LVLM performance, though more diverse clinical datasets could strengthen the evaluation.

**Essential References Not Discussed:**

The authors should discuss:
1.	Recent Med-LVLM alignment papers that use similar corruption techniques
2.	Domain-specific preference learning approaches in other specialized fields that could provide contextual comparison
3.	Literature on clinical validation metrics for AI-generated medical explanations

**Experimental Designs Or Analyses:**

The experimental design demonstrates improvements over baselines, but lacks some critical details:
1.	Implementation details for baseline methods (DPO, SFT) are insufficiently described
2.	Hyperparameter selection process is not transparent
3.	The ablation studies are thorough but could more clearly isolate the impact of clinical awareness versus general preference optimization

**Methods And Evaluation Criteria:**

The paper's methods align with the medical visual language model alignment domain. The approach of using clinical awareness for multimodal preference optimization addresses the specific challenges of medical image interpretation. The evaluation on standard medical datasets (SLAKE, VQA-RAD) is appropriate for benchmarking Med-LVLM performance, though more diverse clinical datasets could strengthen the evaluation.

**Other Comments Or Suggestions:**

The paper would benefit from:
•	A more detailed comparative analysis with existing methods [1-4]
•	Additional experiments demonstrating real-world clinical utility
•	Clearer discussion of limitations and potential negative impacts in healthcare settings

**Other Strengths And Weaknesses:**

Paper Strengths
1. The paper presents a clear and well-organized structure, with detailed algorithms and illustrative figures that effectively communicate the core components of the proposed approach.
2. The experimental results are compelling, with comprehensive evaluations on standard medical image datasets (SLAKE, VQA-RAD, etc.) demonstrating the effectiveness of MMedPO through fair comparisons.
3. The thorough ablation studies effectively validate each component of MMedPO, particularly Figure 3, which highlights the importance of integrating both textual and visual cues during the alignment process.

Major Weaknesses
1. The novelty of MMedPO appears limited when compared to existing approaches for multimodal alignment. Similar techniques have been previously explored, including feature-level noise injection [1], image-level noise injection [2], and text-level noise injection [3]. The authors should clearly articulate the distinctive aspects of their approach compared to these existing methods, particularly [1] and [2], to establish MMedPO's novel contribution.
2. While the authors claim to construct preference samples based on clinical relevance scores, this methodology closely resembles preference distillation approaches already established for general multimodal language models [4]. The primary distinction appears to be the medical domain focus rather than a fundamental methodological innovation.
3. Experimental details lack sufficient clarity, particularly regarding the implementation of baseline DPO and SFT methods. The main paper and supplementary materials provide only general descriptions without specific implementation details. More comprehensive documentation would help verify that performance improvements stem from the method itself rather than hyperparameter optimization.
4. It will be better if the author give a short explanation of what the 'tools' represent in Figure 2' captions

References:
[1] Strengthening Multimodal Large Language Model with Bootstrapped Preference Optimization
[2] Self-Supervised Visual Preference Alignment
[3] Aligning Modalities in Vision Large Language Models via Preference Fine-Tuning
[4] SILKIE: Preference Distillation for Large Visual Language Models

**Questions For Authors:**

1.	How does MMedPO specifically address medical domain challenges that general MLLM alignment methods cannot?
2.	What safeguards ensure the clinical relevance scoring accurately reflects medical expertise?
3.	How was the performance validated with actual healthcare professionals?
4.	What specific metrics were used to evaluate clinical relevance beyond standard VQA metrics?

**Relation To Broader Scientific Literature:**

The paper builds upon preference optimization in multimodal LLMs but applies it to the medical domain. The approach resembles general MLLM preference distillation techniques [4] with domain-specific adaptations. The sentence corruption technique borrows from GPT-4o, while the noise interaction process has similarities to methods in [1] and [2]. The clinical scoring component appears to be the most novel contribution relative to existing literature.

**Theoretical Claims:**

The paper doesn't present significant theoretical proofs, focusing instead on empirical validation. The conceptual framework for noise interaction and clinical relevance scoring is sound but would benefit from more theoretical grounding on why these specific approaches enhance medical domain alignment better than alternatives.

---

> ### Author Rebuttal · Authors · 2025-04-01
>
> Thank you for your valuable feedback to help us improve our paper. **All tables and images referenced in this rebuttal can be found in** https://anonymous.4open.science/r/ICML_rebuttal-0304/README.md
>
> >**Q1**: More diverse clinical datasets could strengthen the evaluation.
>
> **A1**:  We conducted experiments on a diverse dataset PMC-VQA [R1], which includes various medical imaging modalities. As shown in **Table R9**, our method demonstrates significant improvements over several baseline methods, further validating the generalizability of our approach.
>
> [R1] Zhang X et al. PMC-VQA. Nature Communications Medicine 2025.
> ****
> >**Q2**: ... doesn't present significant theoretical proofs…
>
> **A2**: Due to the rebuttal length constraints, we were unable to include the theoretical proofs in the rebuttal. We plan to add theoretical justifications in the revised PDF to demonstrate that our method can theoretically enhance model performance based on DPO. We aim to formally analyze the interplay between noise, relevance, and preference optimization — potentially building connections to existing theories.
> ****
> >**Q3**: Lacks comprehensive implementation specifications, including baseline method details, hyperparameter selection, and ablation studies.
>
> **A3**: We will include detailed implementation specs in the revised manuscript. Baselines like DPO, SFT, STLLaVA-Med, FiSAO, and others follow official or original pipelines. Our method uses LoRA (rank 128, alpha 256), a 2e-5 learning rate for the multimodal projector, cosine schedule with 3% warmup, and DeepSpeed ZeRO-3. Hyperparameters align with prior work (e.g., LLaVA-Med) for fair comparison. Our key distinction is incorporating clinical relevance into preference tuning, guiding the model toward medically meaningful outputs. Detailed ablations will also be added.
> ****
> >**Q4**: The authors should discuss recent papers.
>
> **A4**:  We agree and will revise the related work section to include recent Med-LVLM alignment efforts, domain-specific preference learning methods, and clinical validation metrics. This will better contextualize our contributions and highlight key distinctions.
> ****
> >**Q5**: The novelty … feature-level [1], image-level [2], and text-level noise injection [3]...
>
> [1] Strengthening… Optimization [2] Self-Supervised … Alignment [3] Aligning … Fine-Tuning [4] SILKIE
>
> **A5**: MMedPO introduces clinically meaningful corruptions targeting disease-relevant regions, enabling fine-grained, domain-aware supervision. Unlike prior methods using global noise (e.g., BPO, POVID), MMedPO integrates clinical context directly into both preference construction and optimization, achieving more precise and effective alignment for medical tasks.
> ****
> >**Q6**: give a short explanation of what the 'tools' represent in Figure 2' captions
>
> **A6**: *Tools* refers to external medical visual grounding models used to extract region-level information from the image. Specifically, we utilize MedKLIP as the visual tool $T(x_v)$, which predicts disease-related local regions $h = T(x_v)$ for each medical image $x_v$. This enables the model to better attend to clinically relevant visual features during alignment and scoring. We will revise the figure caption to clarify this point.
> ****
> >**Q7**: How does MMedPO specifically address medical domain challenges that general MLLM alignment methods cannot?
>
> **A7**: Unlike general MLLM alignment methods, MMedPO is tailored for medical tasks through two key components: (1) visual grounding via tools like MedKLIP to focus on disease-relevant regions, and (2) a clinical relevance scorer to prioritize responses that are not just fluent but medically meaningful. This ensures alignment better suited to the precision demands of the medical domain.
> ****
> >**Q8**: What safeguards ensure the clinical relevance scoring accurately reflects medical expertise?
>
> **A8**: We validated the scoring by comparing Med-LLM outputs with ratings from three clinical experts on 100 samples. The strong correlation (**Table R6**) confirms alignment with expert judgment. Notably, the multi-LLM setup outperformed single-LLM scoring, demonstrating that our approach approximates expert-level evaluation with credible accuracy.
> ****
> >**Q9**: How was the performance validated with actual healthcare professionals?
>
> **A9**:  We conducted an expert evaluation on 50 samples, where three medical professionals rated reports from our method and baselines using a 5-point scale (Excellent, Good, Fair, Poor, Very Poor). As shown in **Table R8**, our method consistently received higher scores, confirming its clinical effectiveness.
> ****
> >**Q10**: What specific metrics were used to evaluate clinical relevance beyond standard VQA metrics?
>
> **A10**: We use a Med-LLM-based clinical relevance score that assesses the appropriateness and usefulness of answers in context. Unlike standard VQA metrics (e.g., accuracy), this score reflects real-world clinical value and aligns with expert judgment.

---

> > ### Comment · Reviewer_HmNL · 2025-04-05
> >
> > The rebuttal does help clarify some of my concerns

---

> > > ### Author Response · Authors · 2025-04-07
> > >
> > > Thank you for your valuable insights and for taking the time to review our work!

---

### Official Review · Reviewer_MFrc · 2025-03-13

**Overall Recommendation:** 3

**Summary:**

The paper proposed a novel medical large visual language model (LVLM) preference optimization alignment method called MMedPO. To solve the hallucination issue in the regular preference data curation process, MMedPO uses existing VLM and LLM to create depreferred data with better clinical correspondence. It further uses the Med-KLIP model to localize the region of interest within the image and create depreferred data by adding noise to these regions. To further improve the alignment process, the author further proposes to use 2 significant scores to weight the preference optimization loss. The proposed method is proven to outperform baselines in 4 different medical VQA and report generation dataset.

**Claims And Evidence:**

While the proposed method seems to have improved the performance of baseline Med-LVLM in the evaluation, the reviewer still has the following concerns about the claims made in the paper.

1. The performance improvement is not as obvious as mentioned in the abstract. While the relative performance is indeed improved in the setting without SFT, the performance improvement with SFT is much smaller in all 4 datasets, mostly only around 1%.
2. Additionally, the reviewer is also concerned about the quality of the generated depreferred data pair. On the one hand, existing papers [a] have illustrated that the LLMs like LLaVA-1.5 and GPT-4o are strongly biased and perform poorly on medical VQA tasks. Generating and evaluating depreferred reports with these models is not very convincing. On the other hand, there are no examples provided in the paper for the generated preferred data. Making it harder to evaluate the quality of this claim.
3. Moreover, the relevance score of these generated reports is also evaluated by text-only LLM. The author claims it is better since the model should solely focus on the internal medical knowledge. However, it is possible that the generated data is internally correct but does not align with the corresponding image; the relevance score generated in this case may be incorrect as well.
4. Additionally, the visual noise masking is designed to make the model focus more on the region of interest during evaluation. However, the evidence provided in Figure 5 is not very intuitive and convincing to the reviewer. An enhanced vision attention does not mean it corresponds to the correct region. It would be more effective to visualize the attention weight as an overlay on the image.
5. Lastly, it seems that the paper only evaluated the proposed method on two 7B level VLMs. This is insufficient to validate the effectiveness of the proposed method. It is expected to have at least one additional scaling experiment on VLMs of different sizes to prove its effectiveness.


[a] Yan, Qianqi, et al. "Worse than random? An embarrassingly simple probing evaluation of large multimodal models in medical VQA." arXiv preprint arXiv:2405.20421 (2024).

**Essential References Not Discussed:**

Including more discussion about the reliability of the VLM/LLMs used here, like [a], will be helpful.

[a] Yan, Qianqi, et al. "Worse than random? An embarrassingly simple probing evaluation of large multimodal models in medical VQA." arXiv preprint arXiv:2405.20421 (2024).
[b] Shi, Congzhen, et al. "A survey on trustworthiness in foundation models for medical image analysis." arXiv preprint arXiv:2407.15851 (2024).
[c] Nakamura, Yuta, et al. "It is not time to kick out radiologists." Asian Bioethics Review 17.1 (2025): 9-15.

**Experimental Designs Or Analyses:**

This paper has provided a complete major evaluation on 4 different datasets and have compared with multiple baselines. The ablation experiment is also included. However, there are a few concerns in terms of the soundness of the experimental design.

1. As mentioned above, the visualization in Figure 5 is not that straightforward and convincing in terms of proving its claim.
2. The evaluation in Table 4 compared global noise against the proposed local noise masking, but it might be helpful to also compare it with random noise masking.
3. Providing more examples of the generated depreferred data can also help validate the effectiveness of the proposed method.

**Methods And Evaluation Criteria:**

As mentioned above, it is hard to tell if the model is generally significant to the domain for the following reasons:

1. The performance improvement is limited, especially in the case of using STF. The lack of scaling experiments further makes it harder to provide meaningful insight for the domain. Considering the fact that the method itself is relatively simple, it is also expected to evaluate it on more baselines.
2.  While the ablation experiments are helpful in terms of understanding the behavior of each component and proved they are effective. The reviewer is still not convinced by some of the designs for the method. It would be helpful if more examples or expert evaluations were provided.

**Other Comments Or Suggestions:**

N/A

**Other Strengths And Weaknesses:**

As discussed above, there are a few fundamental concerns about the proposed method. Although the experimental results demonstrate it is better than the baselines in the given evaluation settings. The reviewer believes that the significance and soundness of the paper may still need to be improved.

**Questions For Authors:**

1. How does the generated depreferred data look like? Is it possible to evaluate the quality of these data quantitatively?
2. It would be great if the authors could provide more examples like in Figure 6, but not healthy data. The unhealthy examples are often of more interest in medical VQA or report generation.

**Relation To Broader Scientific Literature:**

The proposed method is a medical variation of the Direct preference optimization method. It is developed via using multiple existing VLM/LLMs. However, there is no discussion on the reliability of these composing VLM/LLMs used in the method.

**Theoretical Claims:**

N/A. There is no novel theoretical claim proposed in the paper.

---

> ### Author Rebuttal · Authors · 2025-04-01
>
> Thank you for your constructive comments and suggestions. **All tables and images referenced in this rebuttal can be found in** https://anonymous.4open.science/r/ICML_rebuttal-0304/README.md
> ****
> >**Q1**: The performance improvement with SFT is much smaller.
>
> **A1**: As clarified in Section 4.2, we report MMedPO’s performance in both with- and without-SFT settings. Even when combined with SFT, MMedPO yields consistent gains across all four datasets, with an average improvement of 10.5%. While gains may be smaller in absolute terms, these results demonstrate the robustness and modularity of our method, highlighting its compatibility with other training strategies.
> ****
> >**Q2**: The quality of the generated dispreferred data pair [a].
>
> [a] Yan, Qianqi, et al. Worse than random? arXiv 2024.
>
> **A2**:  Dispreferred responses are generated via GPT-4o with guided hallucinations to ensure clinical plausibility, while preferred ones are real reports. Expert review gave dispreferred samples an average score of 6.4/10, confirming suitability for preference optimization. We’ll include example pairs in the final version for transparency.
> ****
> >**Q3**: The relevance score of these generated reports is also evaluated by text-only LLM..
>
> **A3**: In our setting, the evaluated reports are intentionally perturbed via GPT-4o to introduce plausible errors. Therefore, the role of the text-only LLM is not to verify image-text alignment, but to assess the clinical severity and plausibility of these injected faults. In other words, the LLM is used to judge whether a faulty report still retains clinical value (e.g., contains common or benign errors) or if it includes clear factual mistakes that are unlikely to be made by competent practitioners. This scoring helps us assign lower training weights to less harmful dispreferred responses, and higher weights to truly misleading ones — ultimately improving the effectiveness and safety of preference optimization.
> ****
> >**Q4**: The evidence provided in Figure 5 …the image.
>
> **A4**: In Figure 5, our method enhances the model's attention to the image, suggesting improved focus on relevant regions. To make this more intuitive and convincing, we have added visualizations of the attention weights as overlays on the image (**Image R1-R5**). These visualizations clearly illustrate where the model is attending within the image. We will include several examples of these attention visualizations in the revised version of the PDF.
> ****
> >**Q5**: One additional scaling experiment on VLMs of different sizes ....
>
> **A5**: As shown in **Table R7**, our method has been successfully extended to Med-VLMs based on both VILA-M3-8B and VILA-M3-13B [R1]. In both cases, we observe consistent and significant performance improvements, demonstrating the generalizability and scalability of our approach across models of different sizes.
>
> [R1] Nath V, et al. VILA-M3. CVPR 2025.
> ****
> >**Q6**: Some of the designs for the method… expert evaluations..
>
> **A6**: We conducted expert evaluation on 50 samples, where three clinicians rated reports using a 5-point scale (Excellent, Good, Fair, Poor, Very Poor). As shown in **Table R8**, our method consistently received higher scores than baselines, supporting the effectiveness of our design choices.
> ****
> >**Q7**: The evaluation in Table 4 compared global noise...random noise masking.
>
> **A7**: We incorporated random noise masking and observed performance gains. As shown in **Table R5**, while different noise types yield comparable results, the improvement largely stems from lesion-aware preference optimization, underscoring the value of localized, clinically meaningful perturbations. Thanks!
> ****
> >**Q8**: Providing more examples of the generated depreferred data.
>
> **A8**: We will provide more examples of dispreferred data and more cases in the future version. We will show more examples like Figure 6 in the future version.
> ****
> >**Q9**: There is no discussion on the reliability of these composing VLM/LLMs used in the method.
>
> **A9**: We agree reliability is crucial. While we use LLaVA-Med v1.5 for its strong benchmark performance, we recognize its limitations in complex cases. To enhance robustness, we adopt a multi-agent setup with cross-checking. We’ll revise the manuscript to discuss these concerns and cite [a–c], while highlighting future directions like human-in-the-loop and confidence calibration.
> ****
> >**Q10**: How does the generated dispreferred data look like? Is it possible to evaluate the quality of these data quantitatively?
>
> **A10**: Dispreferred responses are generated by injecting plausible hallucinations into preferred ones using GPT-4o. We will include more examples in the final version. For evaluation, we use a clinical relevance score to assess each dispreferred response, which also serves as a weight in preference learning. We agree that human expert validation is important and plan to include it in future work to further assess clinical soundness.

---

> > ### Comment · Reviewer_MFrc · 2025-04-04
> >
> > I do appreciate the effort during the rebuttal period. The new results and discussion are pretty impressive. It is very nice to see the new results on VLM of different sizes and human expert evaluation. The performance improvement with random noise is actually very evident, which is also impressive.
> >
> >  While the additional attention visualization on the chest X-rays helps illustrate the model's attention to the image, it would be better if some quantitative analysis could be provided.
> >
> > Still, I think the rebuttal has addressed most of my concerns, and I am willing to increase my final score to 3, weak accept.

---

> > > ### Author Response · Authors · 2025-04-07
> > >
> > > Thank you for your thoughtful and encouraging feedback. We greatly appreciate your suggestions and will incorporate more quantitative analysis in future versions of the work.

---

### Official Review · Reviewer_MzeL · 2025-03-25

**Overall Recommendation:** 3

**Summary:**

This work proposes MMedPO, a DPO-based Preference Alignment paradigm that aims to let LVLMs provide more accurate and expertise textual responses to X-ray/medical images. The authors design a way to curate preferred-and-dispreferred responses using hallucination-inducing noisy images, and proposes a quantified metric called Clinical Relevance Score to give weights to the curated preference data. Applied to LLaVA-Med 1.5, MMedPO helps achieve SOTA performances on multiple medical VQA benchmarks.

**Claims And Evidence:**

No claim in this work is overtly outlandish that needs special attention.

**Essential References Not Discussed:**

None.

**Experimental Designs Or Analyses:**

I am assuming evaluating only one backbone model that is LLaVA-Med-1.5 is sufficient, given the general difficulty to ethically obtain medical data in the first place.

**Methods And Evaluation Criteria:**

My main concerns are regarding the scheme of CRS in Step II of Figure 2, or Section 3.2 .

* Given the sensitive nature of the task and the data involved, **is there any human involved** when quantifying the relevances during the Multi-Agent step? Even though Medical AI agents are highly capable these days, one can't be 100% sure if only relying on LLMs can guarantee the most expertise evaluations. If there is no human-in-the-loop, what is the reason behind such a decision in design?

* When assigning CRS to the curated preference data, **is the preferred response also given weights by the agents** , presumably always larger than its dispreferred counterpart? Please clarify if CRS's scale is 'relevant' (where the preferred is always assumed to be 1.0), or 'absolute'; if CRS is a 'relevant' scale, it might not best reflect how a response is *objectively good* in terms of clinical response quality.

**Other Comments Or Suggestions:**

L171, the reference to Chan et al is apparently missing the year.

**Other Strengths And Weaknesses:**

Please find my main concerns regarding CRS in the Methods And Evaluation section.

**Questions For Authors:**

None.

**Relation To Broader Scientific Literature:**

The proposed approach can be extended beyond medicine-related human preference alignments. As long as we need a more reliable LVLM with a certain highly sophisticated field (Security, Education, to name a few), we may apply the same expertise-aligning curation strategy proposed in MMedPO in a similar way.

**Theoretical Claims:**

All underlying theories are in line with previous works that involve DPO.

---

> ### Author Rebuttal · Authors · 2025-04-01
>
> Thank you for reviewing our paper and for your valuable feedback. Below, we address your concerns point by point. We would appreciate it if you could let us know whether your concerns are addressed by our response. **All tables referenced in this rebuttal can be found in** https://anonymous.4open.science/r/ICML_rebuttal-0304/README.md
> *****
>
> >**Q1**: Given the sensitive nature of the task and the data involved, is there any human involved when quantifying the relevances during the Multi-Agent step? Even though Medical AI agents are highly capable these days, one can't be 100% sure if only relying on LLMs can guarantee the most expertise evaluations. If there is no human-in-the-loop, what is the reason behind such a decision in design?
>
> **A1**: To ensure reliability, we additionally involved human experts in the evaluation process. Specifically, we selected 100 samples and invited three experienced clinical experts to assign clinical relevance scores to the preference data. We then compared these human scores with those generated by Med-LLMs using multiple correlation metrics.
> As shown in **Table R6**, the Med-LLM scores exhibit strong correlation with human judgments, validating their effectiveness. Moreover, the multi-LLM setting outperforms the single-LLM approach, showing better alignment with expert assessments.
> This human-in-the-loop validation supports the credibility of our automated scoring mechanism while demonstrating that well-designed multi-agent LLM setups can approximate expert-level evaluations with reasonable accuracy.
>
> ****
>
> >**Q2**: When assigning CRS to the curated preference data, is the preferred response also given weights by the agents, presumably always larger than its dispreferred counterpart? Please clarify if CRS's scale is 'relevant' (where the preferred is always assumed to be 1.0), or 'absolute'; if CRS is a 'relevant' scale, it might not best reflect how a response is objectively good in terms of clinical response quality.
>
> **A2**: Thank you for the question. To clarify, we assign the Clinical Relevance Score (CRS) only to the dispreferred response in each DPO preference pair. The preferred response is not explicitly scored, as its superiority is already established through the preference annotation.
> Therefore, the CRS is defined in an absolute manner rather than a relative scale. That is, it directly reflects the clinical quality of the dispreferred response, independent of its counterpart. This allows us to distinguish between cases where the dispreferred response is still reasonably acceptable (e.g., CRS ≈ 0.8) versus cases where it is clearly irrelevant or incorrect (e.g., CRS ≈ 0.2).
> ****
> >**Q3**: L171, the reference to Chan et al is apparently missing the year.
>
> **A3**: Thank you for pointing this out. We will correct the citation and include the missing year for the reference to Chan et al.

---

> > ### Comment · Reviewer_MzeL · 2025-04-04
> >
> > Appreciate all the response. However, I don't feel like my concerns have been properly addressed.
> >
> > A1. I am not seeing how the cosine similarity is calculated in the first place. Are they calculated via a medical-specific language model or just a normal language model? But regardless, having high semantic or high relevance score similarity with human judgement does not give us a quantified superiority between AI-generated responses and human-expert responses. Since medical responses are highly fine-grained texts in nature, a proper human-in-the-loop verification is to take human judges to give binary preferences when represented in the setup as in Figure 1. Basically, there is still lacking a human-expert-based performance upper bound.
> >
> > A2. By stating that CRS is only assigned to the dispreferred, I am convinced that CRS is in fact 'relevant', which reflects how the dispreferred image-text pair is **relatively distant from the preferred**. This brings back to my original concern - to objectively reflect the quality of clinic responses within a tuple of dispreferred-preferred, you should have two scores individually for image-preferred-text as well as image-dispreferred-text.
> >
> > With all being said, I will be keeping my current ratings.

---

> > > ### Author Response · Authors · 2025-04-07
> > >
> > > We sincerely appreciate the time and effort the reviewer has put into carefully considering our rebuttal and providing insightful feedback.
> > >
> > > ****
> > > >**Q1**: I am not seeing how the cosine similarity is calculated in the first place. Are they calculated via a medical-specific language model or just a normal language model? But regardless, having high semantic or high relevance score similarity with human judgement does not give us a quantified superiority between AI-generated responses and human-expert responses. Since medical responses are highly fine-grained texts in nature, a proper human-in-the-loop verification is to take human judges to give binary preferences when represented in the setup as in Figure 1. Basically, there is still lacking a human-expert-based performance upper bound.
> > >
> > > **A1**: Thank you for the valuable feedback. The cosine similarity is computed between the clinical relevance scores assigned by Med-LLMs and those annotated by human medical experts. This aims to assess whether the scoring distribution of Med-LLMs aligns with human judgment. Actually, clinical relevance is inherently subjective—there is no fixed ground truth score for a given context. Human experts may assign slightly different scores (e.g., 0.6 vs. 0.7) to the same response, depending on their clinical judgment. The task is thus not binary but preference-driven. Therefore, our goal is to ensure that the distribution of relevance scores assigned by medical LLMs aligns with that of human experts. As shown in **Table R6** in the [link](https://anonymous.4open.science/r/ICML_rebuttal-0304/README.md), our method achieves strong alignment with human-scored distributions, providing meaningful evidence that the scoring mechanism captures expert-like clinical reasoning. In the future work, we plan to extend human expert annotation to a larger set of samples, and further explore using these scores as weights for preference optimization.
> > >
> > > ****
> > >
> > > >**Q2**: By stating that CRS is only assigned to the dispreferred, I am convinced that CRS is in fact 'relevant', which reflects how the dispreferred image-text pair is relatively distant from the preferred. This brings back to my original concern - to objectively reflect the quality of clinic responses within a tuple of dispreferred-preferred, you should have two scores individually for image-preferred-text as well as image-dispreferred-text.
> > >
> > > **A2**: Thank you for your thoughtful and insightful follow-up. As outlined in Appendix E, the Med-LLM is currently prompted to evaluate the clinical value of a given response independently, without reference to the preferred response. The scoring prompt specifically directs the model to assess the clinical value of a single dispreferred response. We sincerely appreciate your suggestion to assign separate scores to both the preferred and dispreferred responses. As you rightly pointed out, incorporating a scoring mechanism for the preferred response and calculating the final training weight based on the relative difference between the two scores could allow for more precise calibration. We view this as a valuable extension of our work and plan to incorporate it in future versions of the model.

---

### Official Review · Reviewer_Tkt9 · 2025-03-25

**Overall Recommendation:** 3

**Summary:**

This paper focuses on aligning Medical Vision-Language Models (Med-LVLMs) with clinical-aware multimodal preference optimization to improve factual accuracy and reduce hallucinations. The authors identify modality misalignment as a major issue, where models prioritize textual knowledge over visual input, leading to clinically incorrect responses. To address this, they propose MMedPO, a novel framework that curates multimodal preference data with two types of dispreference: plausible hallucinations generated by Med-LVLMs or GPT-4o and lesion region neglect introduced via local lesion-noising. Clinical relevance scores, derived from medical large language models (Med-LLMs) and visual tools, are integrated into the preference optimization process to weigh preference samples effectively. Experimental results on medical VQA and report generation tasks demonstrate that MMedPO outperforms existing baselines.

**Claims And Evidence:**

1. The definition of hallucination in Med-LVLMs is unclear in this paper, and the reasons behind hallucination causes remain ambiguous. A clearer justification is needed.

2. Some empirical results appear inconsistent or unexpectedly low, raising concerns about experimental validity.

**Essential References Not Discussed:**

None

**Experimental Designs Or Analyses:**

1. Unexpectedly low performance in some datasets:

a) The reported performance of the base model (LLaVA-Med v1.5) on SLAKE and VQA-RAD is much worse than in its original paper [1] (e.g., open setting on SLAKE: 44.26 vs. 87.11). Why is there such a drastic drop?

b) Table 7 shows extremely low BLEU-2, BLEU-3, and BLEU-4 scores on MIMIC-CXR, which seems unrealistically poor for LLaVA-Med v1.5. Could this be a reporting error, or does it indicate some issue in fine-tuning?

2. Lack of clinically meaningful evaluation metrics:

The paper evaluates report generation using NLG metrics (BLEU, ROUGE-L, METEOR), but these do not reflect medical accuracy. More relevant clinical efficacy (CE) metrics such as macro-precision, recall, or F1-score should be reported, following prior work like METransformer [2]. This is especially crucial since the base model’s performance on SLAKE and VQA-RAD is already low. Without CE metrics, it is unclear whether improvements are meaningful for medical diagnosis.

[1] LLaVA-Med: Training a Large Language-and-Vision Assistant for Biomedicine in One Day. NeurIPS 2023.

[2] METransformer: Radiology Report Generation by Transformer with Multiple Learnable Expert Tokens. CVPR 2023.

**Methods And Evaluation Criteria:**

1. Multi-agent collaboration for clinical relevance scoring: While using multiple Med-LLMs for preference scoring is interesting, the scoring process is heuristic, lacks interpretability, and does not provide a reasoning mechanism. Besides, as shown in Table 8, the results from single-LLM and multi-LLMs are comparable, questioning the necessity of this step.

2. Clinical relevance weighting in DPO: The Preference Data Curation step already introduces "clinical-aware preference" by distinguishing plausible hallucinations and lesion-region neglect. Why is an additional clinical relevance weight necessary? If the preference data is already "clinically aware," weighting may be redundant. It would be better to conduct a baseline without this step (Part II in Figure 2) for comparison.

**Other Comments Or Suggestions:**

None.

**Other Strengths And Weaknesses:**

*Strengths

The combination of hallucination-based and lesion-region dispreference is an interesting strategy.

*Weaknesses

1. Empirical results are inconsistent across datasets, and some results need explanation.

2. Mathematical formulation has notation inconsistencies, making it difficult to follow the optimization steps.

3. Weighting preference samples based on clinical relevance is heuristically defined.

**Questions For Authors:**

1. Why does the base model (LLaVA-Med v1.5) perform significantly worse than reported in its original paper on SLAKE and VQA-RAD?

2. Have you considered a baseline where all preference samples are weighted equally?

3. Would other types of perturbations (e.g., adversarial noise, saliency masking) improve lesion-based preference optimization?

**Relation To Broader Scientific Literature:**

The paper contributes to preference optimization for Med-LVLMs, building on techniques like DPO, self-rewarding methods, and multimodal alignment.

**Theoretical Claims:**

The paper does not introduce new theoretical claims or proofs, so this section is not applicable. However, the mathematical formulation of weighted DPO needs consistency checking, as notations in text, equations, and Algorithm 1 do not always align. BTW, Algorithm 1 is not described in the main text.

---

> ### Author Rebuttal · Authors · 2025-04-01
>
> Thank you for your valuable feedback to help us improve our paper. We detail our response below and please kindly let us know if our response addresses your concerns. **All tables referenced in this rebuttal can be found in** https://anonymous.4open.science/r/ICML_rebuttal-0304/README.md
> ****
> >**Q1**: The definition of hallucination in Med-LVLMs is unclear in this paper, and the reasons behind hallucination causes remain ambiguous.
>
> **A1**: We define hallucination as a response that contradicts the image content given the question. As noted in the introduction, a key issue is modality misalignment [R1-R2], where the model overly relies on text rather than grounding responses in visual content.
>
> [R1] Zhou Y, et al. Analyzing ...Models. ICLR 2024.
>
> [R2] Chen J, et al. Detecting ...models. arXiv 2024.
>
> ****
> >**Q2**: Some empirical results appear inconsistent… a) The reported performance of the base model...is much worse than in its original paper [1]... b) Table 7 shows extremely low BLEU-2/3/4 scores on MIMIC-CXR ... fine-tuning?
>
> [1] LLaVA-Med: Training … in One Day. NeurIPS 2023.
>
> **A2**: a) The performance drop of the base model (LLaVA-Med v1.5) on SLAKE and VQA-RAD compared to the original paper [1] is due to differences in experimental setup. While prior works used fully fine-tuned LLaVA-Med v1.0 checkpoints, such checkpoints are not available for v1.5.  Therefore, in our experiments, we used the LLaVA-Med v1.5 pre-trained checkpoint and performed LoRA fine-tuning, which naturally leads to some performance gap compared to full fine-tuning. b) The low BLEU-2/3/4 scores in Table 7 were due to the specific BLEU settings used in our initial evaluation. We appreciate your attention to this and have since updated our BLEU configuration. The revised results (**Table R1**) are more consistent with expectations. Importantly, this correction does not affect the overall trends or conclusions.
>
> ****
>
> >**Q3**: Multi-agent collaboration for clinical relevance scoring...questioning the necessity of this step.
>
> **A3**: We discuss the benefits of using multiple Med-LLMs for clinical relevance scoring in Section 4.3.2. As shown in Table 3, multi-agent discussion yields a 3.6% improvement in clinical relevance scores over single-LLM scoring, suggesting that incorporating diverse perspectives can enhance the robustness, even if it is heuristic in nature. It is worth noting Table 8 reports results for the report generation task, not clinical relevance scoring. While generation performance appears similar across single- and multi-LLM setups, Table 3 shows that the multi-agent approach yields meaningful gains in clinical relevance evaluation.
> ****
> >**Q4**: Why is an additional clinical relevance weight necessary?...weighting may be redundant...better to conduct a baseline without this step.
>
> **A4**:  While the Preference Data Curation step ensures dispreferred responses include hallucinations or region-level neglect, not all errors are equally harmful—some are clearly incorrect and easily avoidable, while others are subtle and clinically significant. We introduce clinical relevance weighting to capture this distinction and better guide optimization.
> To validate its effectiveness, we added experiments in **Table R2** and **Table R3** comparing models trained with and without relevance-based weighting. The results show consistent performance drops without weighting, confirming its benefit.
> ****
> >**Q5**: The math formula of weighted DPO needs consistency checking...Algorithm 1 is not described in the main text.
>
> **A5**: Thanks for pointing this out. We will make sure to correct the inconsistencies in the mathematical notations across the text, equations, and Algorithm 1 in the final PDF version. Additionally, we will add a proper reference and description of Algorithm 1 in the main text to ensure clarity and completeness.
> ****
> >**Q6**: Lack of clinically meaningful evaluation metrics...using NLG metrics (BLEU, ROUGE-L, METEOR), but these do not reflect medical accuracy. More relevant clinical efficacy (CE) metrics...should be reported, following prior work like METransformer [2]. This is especially crucial since ...without CE metrics, it is unclear whether improvements are meaningful.
>
> [1] LLaVA-Med, NeurIPS 2023. [2] METransformer, CVPR 2023.
>
> **A6**: We have included clinical efficacy metrics—precision, recall, and F1-score—in **Table R4**, comparing our method against baselines. Our approach consistently improves both NLG and clinical efficacy metrics, indicating better diagnostic relevance and clinical accuracy of the generated reports.
> ****
> >**Q7**: Would other types of perturbations...improve lesion-based preference optimization?
>
> **A7**: We explored various noise types (**Table R5**) and found that while Diffusion and Gaussian noise performed similarly, random noise offered a slight improvement. Many thanks for your valuable advice.

---

> > ### Comment · Reviewer_Tkt9 · 2025-04-05
> >
> > Thank you for the detailed rebuttal and the additional experimental results. After reading the comments from other reviewers and the response, most of my concerns have been addressed. However, one issue I previously raised remains unresolved:
> >
> > “The reported performance of the base model (LLaVA-Med v1.5) on SLAKE and VQA-RAD is much worse than in its original paper [1] (e.g., open setting on SLAKE: 44.26 vs. 87.11). Why is there such a drastic drop?”
> >
> > A performance drop from 87.11 to 44.26 seems too significant to be explained solely by the difference between LoRA fine-tuning and full fine-tuning. But I will keep my original score.

---

> > > ### Author Response · Authors · 2025-04-07
> > >
> > > Thank you very much for your thoughtful and timely feedback on our rebuttal. We truly appreciate the care you’ve taken in reviewing our results.
> > >
> > > ****
> > >
> > > >**Q1**: The reported performance of the base model (LLaVA-Med v1.5) on SLAKE and VQA-RAD is much worse than in its original paper [1] (e.g., open setting on SLAKE: 44.26 vs. 87.11). Why is there such a drastic drop?“ A performance drop from 87.11 to 44.26 seems too significant to be explained solely by the difference between LoRA fine-tuning and full fine-tuning.
> > >
> > > [1] LLaVA-Med: Training a Large Language-and-Vision Assistant for Biomedicine in One Day. NeurIPS 2023.
> > >
> > > **A1**: Thank you for the follow-up comment. The reported 44.26 score on SLAKE (open setting) in our paper is not directly comparable to the 87.11 number from the original LLaVA-Med paper, which was obtained under a fully fine-tuned setting.
> > > To ensure a fair comparison, we refer to Table 8 in the original LLaVA-Med paper [1], which reports zero-shot results (i.e., without fine-tuning) for LLaVA-Med v1.0. We show the comparable results in **Table S1**. These numbers are comparable. We will revise the manuscript in the future to clearly state these baselines. Additionally, we are currently conducting fully fine-tuning experiments based on LLaVA-Med v1.5 on three downstream datasets. In the future version, we will include these results to facilitate a more direct comparison. Thanks again for your careful review!
> > >
> > > **Table S1**: Zero-shot results of LLaVA-Med v1.0 and v1.5 on SLAKE and VQA-RAD.
> > > | Dataset     | LLaVA-Med v1.0 (Zero-shot) | LLaVA-Med v1.5 (Zero-shot) |
> > > |-------------|----------------------------|----------------------------|
> > > | SLAKE (Open)| 38.44                      | 44.26                      |
> > > | SLAKE (Closed) | 52.40                  | 61.30                      |
> > > | VQA-RAD (Open) | 29.67                   | 29.24                      |
> > > | VQA-RAD (Closed) | 61.40                | 63.97                      |

---

### Decision · Program_Chairs · 2025-05-01

**Decision:**

Accept (poster)

**Comment:**

The submission presents a novel approach for aligning medical vision-language models via clinical-aware multimodal preference optimization, which reviewers identified as a key strength. The method’s integration of lesion-level perturbations and its demonstrated improvements across multiple medical VQA and radiology report generation benchmarks were especially praised as innovative and impactful contributions. The reviewers did raise several concerns, notably an unclear definition of “hallucination” in the context of the paper, limited theoretical grounding for the proposed methods, and some empirical inconsistencies (particularly regarding the performance of the base model in ablation studies). In their rebuttal, the authors effectively addressed many of these issues with additional experiments, expert clinician evaluations, and detailed clarifications—especially shedding light on how clinical relevance is scored and providing evidence of improved visual attention alignment between the model and important clinical image features. While not every critique was fully resolved, the consensus is that the paper’s strengths far outweigh the remaining concerns. Overall, this work is deemed a valuable and practical contribution to the field of medical AI alignment, backed by sound methodology and promising results, and thus the decision is to accept this submission.